# From Acute Infection to Prolonged Health Consequences: Understanding Health Disparities and Economic Implications in Long COVID Worldwide

**DOI:** 10.3390/ijerph21030325

**Published:** 2024-03-11

**Authors:** Jaleel Jerry G. Sweis, Fatima Alnaimat, Valeria Esparza, Supritha Prasad, Abeera Azam, Zeel Modi, Mina Al-Awqati, Pim Jetanalin, Nadia J. Sweis, Christian Ascoli, Richard M. Novak, Israel Rubinstein, Ilias C. Papanikolaou, Nadera Sweiss

**Affiliations:** 1Division of Cardiology, Department of Medicine, University of Illinois Chicago, Chicago, IL 60612, USA; jsweis20@uic.edu; 2Department of Internal Medicine, Division of Rheumatology, School of Medicine, University of Jordan, Amman 11942, Jordan; 3Department of Medicine, University of Illinois Chicago, Chicago, IL 60612, USA; vespar2@uic.edu (V.E.); spprasa2@uic.edu (S.P.); zmodi2@uic.edu (Z.M.); 4Department of Internal Medicine, The University of Texas Health Science Center at Tyler, Tyler, TX 75701, USA; abeera.azam@uttyler.edu; 5Division of Rheumatology, Department of Medicine, University of Illinois Chicago, Chicago, IL 60612, USA; alawqati@uic.edu (M.A.-A.); pimlin@uic.edu (P.J.); nsweiss@uic.edu (N.S.); 6Department of Business Administration, King Talal School of Business Technology, Princess Sumaya University for Technology, Amman 11942, Jordan; n.sweis@psut.edu.jo; 7Department of Medicine, Division of Pulmonary Critical Care Sleep and Allergy, University of Illinois Chicago, Chicago, IL 60612, USA; cascoli@uic.edu (C.A.); irubinst@uic.edu (I.R.); 8Division of Infectious Diseases, University of Illinois, Chicago, IL 60612, USA; rmnovak@uic.edu; 9Department of Respiratory Medicine, Sarcoidosis Clinic, Corfu General Hospital, 49100 Corfu, Greece; iliaspapanikolaou@gnkerkyras.gr; 10Department of Internal Medicine, School of Medicine, University of Jordan, Amman 11942, Jordan

**Keywords:** long COVID-19, fatigue, health disparities, economics

## Abstract

The COVID-19 pandemic has resulted in a growing number of patients experiencing persistent symptoms and physiological changes after recovering from acute SARS-CoV-2 infection, known as Long COVID. Long COVID is characterized by recurring symptoms and inflammation across multiple organ systems. Diagnosis can be challenging, influenced by factors like demographics, comorbidities, and immune responses. Long COVID impacts various organ systems and can have neuropsychological effects. Health disparities, particularly related to race, contribute to a higher burden of infection and ongoing symptoms in minority populations. Managing Long COVID entails addressing a spectrum of symptoms that encompass physical, cognitive, and psychological aspects. The recovery period for patients with Long COVID can vary significantly, influenced by factors like the severity of the disease, hospitalization, comorbidities, and age. Currently, there are no universally effective treatments, although certain interventions show promise, necessitating further research. Self-management and rehabilitation programs can provide relief, but more research is needed to establish their effectiveness. Preventive measures such as vaccination and the use of antiviral medications and metformin. It is imperative to conduct further research to develop evidence-based guidelines and gain a better understanding of the long-term implications of COVID-19. Long COVID could have substantial economic impact on the labor market, productivity, healthcare expenditures, and overall economic growth. To address the challenges patients with long-term complications face, there is a focus on strategies like promoting telework and flexible work arrangements to accommodate diverse symptoms, particularly chronic fatigue and other Long COVID effects. In conclusion, this review emphasizes the multifaceted complexity of Long COVID and the ongoing need to address its potential long-term health and economic impacts.

## 1. Introduction

A substantial number of patients with COVID-19 continue to experience lingering symptoms. Studies estimate that approximately 1 out of 5–8 people who have had COVID-19 may develop Long COVID-19 [1]. Observational evidence suggests that these patients may experience a wide range of symptoms following their recovery from the acute phase of the illness. Different terms have been used to describe this condition, such as “Long COVID”, “post-COVID conditions”, “post-acute sequelae of SARS-CoV-2 infection”, “post-acute COVID-19”, “chronic COVID-19”, and “post-COVID syndrome” [2] For this review the term Long COVID will be used. While certain aspects of this syndrome may be distinct to COVID-19, many of them appear to resemble the recovery process observed in other viral illnesses, critical illnesses, and/or sepsis [3,4,5,6]. Long COVID can be described as a diverse clinical and physiological condition characterized by persistent or recurring symptoms and physiological changes [7]. These symptoms primarily indicate an inflammatory disorder that emerges in some patients following the resolution of SARS-CoV-2 infection, in the absence of other infections or associated conditions. Consequently, long COVID can be regarded as a complex chronic immunoinflammatory disorder, driven by a hyperinflammatory state [8]. Although there is currently no universally accepted definition for this syndrome, its clinical and physiological presentation serves as its primary characterization. Further refinement and clarification are necessary to establish a comprehensive understanding of Long COVID [8]. Importantly, the pandemic has highlighted both the severe health impacts of the virus and existing health disparities among different racial and ethnic groups [9]. As the world confronts the challenges of long COVID, it becomes crucial to explore the intersection of race with this condition and its global implications.

This review delves into the clinical manifestations, diagnostic criteria, underlying pathophysiology, long-term organ consequences, complications, risk factors, and therapeutic interventions in patients with Long COVID. In addition, we will address its economic ramifications worldwide.

## 2. Methods

Following published guidance on writing narrative reviews [10], our literature search methodology involved a targeted exploration of published articles focusing on long COVID-19, with a cutoff date of 30 July 2023. The keywords utilized included “COVID”, “Long COVID”, “post-COVID conditions”, “post-acute sequelae of SARS-CoV-2 infection”, “post-acute COVID-19”, “chronic COVID-19”, and “post-COVID syndrome”. Review articles, original studies, and case reports were included, while book chapters were not considered. Article selection was at the discretion of the authors responsible for writing the assigned subsections based on their expertise. We searched across diverse online platforms, including MEDLINE, Scopus, Web of Science, and the Google Scholar search engine, ensuring a comprehensive data collection. The articles used in writing this review were cited, and the full list of references is available at the end of the manuscript.

## 3. Clinical Manifestations of Long COVID: Immunological and Pathophysiological Mechanisms

According to the World Health Organization (WHO), Long COVID is characterized by symptoms that persist or emerge three months after the initial COVID-19 infection and last for at least two months, with no other diagnosis explaining these symptoms [11]. Despite being primarily recognized as a respiratory illness, COVID-19 can impact almost every organ system. Accordingly, Long COVID exhibits a broad spectrum of extrapulmonary sequelae, affecting various body organs [12]. The number of symptoms may vary, and there is no specific requirement for the number of symptoms needed for diagnosis.

Commonly reported symptoms include:FatigueRespiratory symptomsDifficulty sleepingMusculoskeletal pain/weaknessCognitive dysfunction

These symptoms generally interfere with the daily functioning of the patient [3,4,13,14].

The pathophysiology of Long COVID involves several proposed mechanisms, each contributing to the described symptoms:Cell Damage: Viral invasion directly damages cells, potentially causing respiratory symptoms due to lung tissue damage [15,16]Immune System Dysregulation: Long COVID is linked to immune system dysregulation. Reduced CD4+ and CD8+ T cells, along with increased interferon expression, may lead to various symptoms [17]Inflammation: In response to the virus, the body initiates inflammation. This can result in musculoskeletal pain, weakness, and other symptoms [15,16].Organ Tissue Injury: Severe illness from the virus can lead to organ tissue injury, contributing to multiple symptoms. Cognitive dysfunction may arise from brain tissue damage [16]

Furthermore, Long COVID is associated with elevated autoantibody levels, including those targeting angiotensin-converting enzyme receptor 2 (ACE2) [18,19]. These antibodies emerge post-SARS-CoV-2 infection, potentially intensifying the immune system’s response and leading to persistent inflammation. Arthur et al. found that ACE2 antibodies were detected in a significant proportion of patients who had previously been infected with SARS-CoV-2, and these antibodies were associated with lower soluble ACE2 activity in plasma. Consequently, decreased ACE2 levels after COVID-19 infection leads to elevated Ang II levels and a heightened immune system response, potentially causing persistent inflammation [20].

### 3.1. Musculoskeletal System Involvement

Fatigue is the most common symptom reported by patients experiencing Long COVID, whether hospitalized or not for the COVID-19 disease [21].Subramanian et al. (2022) evaluated 486,149 non-hospitalized patients with SARS-CoV-2 infection and identified three major clusters of Long COVID phenotypes [22]:
Broad Spectrum of symptoms: This is the most common cluster and includes a wide range of symptoms, such as pain, fatigue, and rash (80%).Respiratory Symptoms: This is the least common and mainly involves persistent respiratory symptoms (cough, shortness of breath and phlegm (5.8%))Mental health and cognitive symptoms, including anxiety, depression, insomnia and brain fog (14.2%).
Joint pain and myalgias are thought to be manifestations of the proinflammatory effects of the viral infection and modified expression of neuromuscular endogenous markers [23]. In normal physiology, myokines are released during physical activity and induce an anti-inflammatory state in the body by reducing the number of macrophages subtype 1 (M1—pro-inflammatory) and increasing the number of macrophages subtype 2 (M2—anti-inflammatory) [24]. However, in the presence of SARS-CoV-2, there is an induction of the opposite physiologic state, increasing the rate of M1, which stimulates an increase in IL-1, TNF-α and Toll-like receptors (TLRs) resulting in a prolonged muscular inflammatory environment [23,24,25].Furthermore, corticosteroids utilized in treating acute COVID-19 can lead to musculoskeletal complications like myopathy, avascular necrosis, and osteoporosis fractures [26].

### 3.2. Respiratory System

Respiratory Symptoms in Long COVID:Patients with Long COVID have a twofold increased risk of developing respiratory conditions [27]. The most frequent respiratory symptoms experienced after acute COVID-19 include dyspnea and cough [5,28,29,30].Long-term pulmonary complications are common and may involve:
Ventilator Dependence: Some patients may require ongoing ventilator support or supplemental oxygen.Tracheostomy: In more severe cases, a tracheostomy might be necessary.Abnormal pulmonary function (PFTs): Pulmonary function tests (PFTs), which measure lung function, often reveal abnormalities. These abnormalities may affect Forced Expiratory Volume 1 (FEV1), which measures how much air you can exhale in one second, Forced Vital Capacity (FVC), the total amount of air exhaled, and Total Lung Capacity (TLC), the maximum amount of air the lungs can hold [31].Fibrotic Lung Disease: In some cases, patients may develop fibrotic lung disease, characterized by scarring of lung tissue [30].In a large multi-center study [32], around one-third of patients who underwent PFT six months following discharge had an abnormal FEV1, FVC, or TLC [31]. At the same time, another study reported Diffusing Capacity of Lungs for Carbon monoxide (DLCO) as the most common lung function abnormality. Ground glass opacity was the most common high-resolution computed tomography (CT) pattern observed six months after hospital discharge [13,33]. Over 40% of patients with chest CT scans during acute COVID-19 infection showed abnormal findings at the 3-month follow-up. Factors such as higher severity scores on initial CT scans, more comorbid conditions, longer hospital stays, and increased ICU admission rates were associated with this risk [34].As seen in pulmonary function testing and imaging, COVID-19 infection can lead to lung fibrosis, restrictive abnormalities, airway obstruction, and reduced diffusion capacity [35]. Histopathologic examination of post-mortem cases has also revealed fibroproliferative changes [16], which are believed to result from systemic inflammatory responses. Prolonged exposure to supplemental oxygen can induce oxidative stress, triggering inflammation and eventually leading to lung fibrosis [36].Furthermore, COVID-19 infection is associated with a hypercoagulable state characterized by complex coagulation activation and thrombotic microangiopathy. Over time, partially resolved blood clots with restricted blood flow may contribute to ongoing respiratory symptoms [29,37].

### 3.3. Cardiovascular System

Cardiovascular Symptoms in Long COVID:

Patients with Long COVID often report symptoms such as palpitations, chest pain, tachycardia, and orthostatic intolerance [29]. In a survey by Ziauddeen et al., out of 2550 patients, 89% reported cardiopulmonary symptoms related to Long COVID [38].

Post-acute cardiovascular (CV) abnormalities associated with Long COVID include:
Myocardial infarction (the most prominent long-term effect [39]Right ventricular dysfunctionViral myocarditisAutoimmune dysfunctionArrhythmiasAn observational study of 100 patients a few months after the onset of COVID-19 revealed that more than three-quarters of patients showed abnormal cardiovascular magnetic resonance (CMR) imaging at 2–3 months (median duration 71 days) after their positive COVID-19 test [33]. The most common abnormalities observed were myocardial inflammation, regional scar, and pericardial enhancement.In a prospective study of patients hospitalized with severe COVID-19 pneumonia and myocardial injury [39], repeat echocardiograms at three months showed persistent adverse ventricular remodeling, particularly right ventricular dilatation, and dysfunction in nearly one-third of the patients.Long COVID’s cardiovascular effects have been linked to various mechanisms:Viral cardiomyocyte invasion: The virus invades cardiomyocytes through angiotensin-converting enzyme 2 receptors, leading to direct cytotoxic and inflammatory effects [16]. Autopsy studies have revealed evidence of myocarditis, lymphocytic infiltration, and myocyte necrosis, with viral particles found in most cases [39].Myocardial injury and fibrosis: Elevated troponin levels in Long COVID cases indicate myocardial injury [33,35], which may activate fibrosis pathways and contribute to cardiac remodeling. As a result, patients may experience heart failure or arrhythmia [16].Autoimmune and thrombotic complications: Autoimmune responses targeting cardiac antigens, possibly through molecular mimicry, have been implicated in vascular and thrombotic complications in Long COVID [33,35]. Increased risk of thromboembolic events has been reported in several studies, mostly in patients who required critical care admission [40].

### 3.4. Nervous System

“Neuro-COVID” or “Post-COVID-19 Neurologic Syndrome”, both used interchangeably, refers to a series of symptoms that include headaches, fatigue, cognitive impairment, anxiety/depression, insomnia, alteration of smell and taste, and vertigo [29,41]. Neurologic events such as stroke, encephalitis, or neuronal injury from hypoxia or hypotension during acute COVID-19 infection may have a lasting influence on an patient’s daily functioning, even months to years following the infection [42]. Additionally, cases of COVID-19 resulting in delirium, generally in severe cases requiring ICU level of care or prolonged hospitalization, have been suggested to carry a greater risk of neurological symptoms of Long COVID [43].

A retrospective study compared whole-brain positron emission tomography (PET) scans of 35 patients with a history of COVID-19 infection and current symptoms of fatigue and neurological complaints to 44 healthy subjects [44]. Patients with Long COVID exhibited hypometabolism in multiple brain regions, and these hypometabolic clusters served as highly discriminant markers between Long COVID patients and healthy subjects. The areas of hypometabolism also showed correlations with specific symptoms, including hyposmia/anosmia, memory/cognitive impairment, pain, and insomnia. These areas of hypoperfusion were consistent with associations previously described in the literature, such as cerebellar hypoperfusion associated with anosmia [45] and additional frontal cortex and brainstem hypoperfusion related to chronic pain [46].

Neurological symptoms in Long COVID, including headaches, cognitive changes (“brain fog”), and smell and taste disturbances, are believed to be associated with persistent neuroinflammation, microglia activation, and micro-thromboses. Post-mortem autopsy studies and high-resolution magnetic resonance imaging of the brain have provided evidence of microvascular injury and perivascular-activated microglia [42]. Fatigue, another common neurological sequela of COVID-19, is likely a multifactorial consequence involving psychosocial factors, oxidative stress leading to mitochondrial dysfunction in muscles, and accumulation of toxic substances in the nervous system [47].

### 3.5. Renal System Involvement

One-third of previously hospitalized COVID-19 patients with acute kidney injury demonstrated adverse renal outcomes [48]. SARS-CoV-2 enters human cells through the ACE2, which is widely distributed throughout the body [20,49]. In essence, ACE2 functions as a critical counterpart to ACE [50]. ACE’s primary role is to convert the vasodilator angiotensin I into the vasoconstrictor angiotensin II. ACE2, on the other hand, has a broader function, which includes breaking down various peptide hormones. Its most vital role is removing a specific amino acid, phenylalanine, from angiotensin II, effectively converting it back into angiotensin. ACE2 is also involved in cleaving other peptides such as bradykinin, apelin, neurotensin, dynorphin A, and ghrelin. Significantly, ACE2 receptors act as gateways for certain coronaviruses, including HoV-NL63, SARS-CoV, and SARS-CoV-2, to enter human cells [51]. In the kidney, podocytes, and cells from the proximal tubule express ACE2 receptors on their surface, enabling SARS-CoV-2 invasion [15]. The infection leads to the downregulation of the ACE2 receptor, increasing angiotensin II levels, which, through its action on the AT1 receptor, causes systemic injury [52].

Various mechanisms have been presumed to cause acute and chronic kidney injury after the acute phase of COVID-19 infection:Endothelial and Podocyte Damage: COVID-19 can lead to damage of the endothelial cells and podocytes within the kidney. This endothelial injury can disrupt the filtration function of the kidneys and, over the long term, result in impaired kidney function. Additionally, podocyte damage can lead to proteinuria (excess protein in the urine), which is a known risk factor for chronic kidney disease [53].Cytokine Release and Complement Activation: The immune system’s response to COVID-19 involves the release of various cytokines and the activation of the complement system. Prolonged or excessive activation of these processes can have harmful effects on kidney tissues. Understanding how these immune responses relate to kidney injury is crucial for predicting and managing long-term kidney function [53].Microthrombi in Renal Circulation: The presence of microclots (microthrombi) within the blood vessels of the kidney can impair blood flow and contribute to kidney damage. Over time, this can lead to chronic kidney disease, emphasizing the importance of monitoring and managing kidney health in COVID-19 survivors [54,55].COVID-19-Associated Nephropathy (COVAN): COVAN is a severe form of kidney injury directly related to COVID-19. It disproportionately affects patients with specific genetic traits, such as the Apolipoprotein L1(APOL1) gene polymorphism, particularly those of African descent. Understanding the genetic and racial factors associated with COVAN can help identify at-risk populations and develop tailored treatment and prevention strategies [55,56].

The impact of COVID-19 on kidney function remains uncertain. Future research should include extensive prospective studies with extended follow-up periods. These studies should comprehensively evaluate kidney involvement through various means, including kidney biopsies, urinalysis, serum creatinine and cystatin C measurements, direct measurement of glomerular filtration rate, and assessment of tubular function through urinary β2-microglobulin measurements.

### 3.6. Gastrointestinal System Involvement

A systematic review of 50 studies found gastrointestinal symptoms in approximately 12% of Long COVID patients [57]. Prolonged shedding of viral particles from the GI tract could be responsible for some of the common gastrointestinal manifestations of Long COVID such as chronic abdominal pain, loss of appetite, nausea and vomiting [24,57]. Another potential reason may be attributed to potential alterations in the gut microbiome, impacting the symbiotic microorganisms within the gastrointestinal (GI) tract. The human microbiota constitutes a diverse microbial community that resides harmoniously in various anatomical sites throughout the body. This microbial community within the gut, known as the gut microbiota, plays a pivotal role in various functions, including food fermentation, vitamin synthesis, immune system maturation, and defense against harmful microorganisms. As a result, it is crucial for maintaining overall well-being. These changes in the gut microbiome could potentially increase patient susceptibility to viral antigens and opportunistic infections in the GI tract [57,58,59,60].

### 3.7. Endocrine System Involvement

Endocrine complications of Long COVID primarily involve the pancreas and thyroid gland. The pathophysiology is not conclusive; however, it appears to be related to direct viral injury, immunological and inflammatory damage or iatrogenic complications [61,62,63]. The virus may cause damage to pancreatic B-cells through ACE2 expression, leading to transient diabetes during acute SARS-CoV-2 infection. In the long term, cases of worsening type 2 diabetes mellitus and ketoacidosis have been observed [61,62]. Y Xie and Al-Aly analyzed 181,280 participants with a positive COVID-19 test and two control groups using the US Department of Veterans Affairs databases [64]. They found that COVID-19 survivors faced increased risks and burdens of incident diabetes and antihyperglycemic use beyond the first 30 days of infection. In the thyroid gland, viral invasion and cytokine release induce inflammation, potentially resulting in hypothyroidism that may sometimes become permanent [63].

### 3.8. Psychological Symptoms/Sequelae

Long COVID patients commonly experience chronic fatigue, anxiety, depression, and post-traumatic stress disorder [5,41]. Furthermore, COVID-19 can also lead to significant neuropsychiatric diseases such as encephalitis, seizures, and cognitive impairment [65]. A systematic review involving 1,285,407 participants from 32 countries revealed that 20% of COVID-19 survivors exhibited psychiatric symptoms during the 12 months after recovery, which is consistent with rates observed in survivors of other coronaviruses like severe acute respiratory syndrome coronavirus (SARS) and Middle East Respiratory Syndrome (MERS) [66]. Among hospitalized COVID-19 patients, approximately 42% reported some post-traumatic stress disorder (PTSD) symptoms at 3–4 months after discharge [67].

### 3.9. Cutaneous Symptoms/Sequelae

An international registry for COVID-19 dermatological manifestations, established via a collaboration between the International League of Dermatological Societies and the American Academy of Dermatology, reported a total number of 716 cases of new-onset dermatologic signs and symptoms in patients with confirmed or suspected COVID-19. Common skin findings in COVID-19 patients included the following:Morbilliform: This term refers to a rash that resembles measles, characterized by small, red, and slightly raised spots on the skin.Pernio-like: Pernio is a medical term for chilblains, which are inflammatory skin lesions that can occur in response to cold temperatures. Pernio-like means skin findings resembling chilblains.Urticarial: Urticaria is the clinical term for hives, which are raised, itchy welts on the skin often caused by an allergic reaction.Macular erythema: This describes redness of the skin without any raised bumps or blisters. Macular means flat, and erythema refers to redness.Vesicular: Vesicles are small fluid-filled blisters on the skin.Papulosquamous: This term combines ‘papules’ (small, raised bumps) and ‘squamous’ (scaly or flaky) and is used to describe skin conditions with these characteristics.Retiform purpura: Purpura refers to purple or red discoloration of the skin caused by bleeding under the skin. ‘Retiform’ might not be a common term, but it can be explained as a network-like pattern of purpura, often seen in certain skin conditions.

These various skin findings have been observed in patients with confirmed or suspected COVID-19 [68].

Furthermore, approximately 28.6% of COVID-19 survivors experienced alopecia three months after hospital discharge, with a higher prevalence in women (48.5%). Most cases of alopecia started after discharge (72%) [4]. Some skin manifestations, such as livedo reticularis, acral lesions, morbilliform, and urticarial rashes, have persisted after SARS-CoV-2 infection, but the underlying pathophysiology remains poorly understood [69].

## 4. Risk Factors and Demographics Leading to Increased Susceptibility for Long COVID

Several factors are associated with an increased risk of developing Long COVID. These factors encompass demographic characteristics, comorbidities, immunological response, and COVID-19 severity.

**1.** 
**Demographic Characteristics:**
Female sex, belonging to an ethnic minority, smoking, BMI and experiencing socioeconomic deprivation have been linked to a higher risk of Long COVID [22,70,71,72].
**2.** 
**Comorbidities:**
Conditions like anxiety, depression, chronic obstructive pulmonary disease, and fibromyalgia contribute to an increased risk of Long COVID [22].
**3.** 
**Age:**
Studies on the relationship between age and Long COVID have yielded conflicting results, leading to a lack of consensus among researchers [22,71,72].
**4.** 
**COVID-19 Severity and Vaccination Status:**
Patients who have not received the COVID-19 vaccine or have experienced severe cases requiring hospitalization or admission to the intensive care unit are believed to be at an elevated risk of developing Long COVID [70,73].Hypothetically, we may postulate that a heightened risk of developing Long COVID, which includes symptoms like brain fog, is associated with the severity of COVID-19 for several reasons. Firstly, severe COVID-19 often triggers a robust immune response and cytokine storm, leading to more extensive organ damage, potentially affecting the brain. Secondly, the treatment for severe COVID-19 is often more aggressive and may carry a higher risk of iatrogenic harm, such as complications from intubation or nosocomial infections, which can result in enduring consequences. Furthermore, patients who initially presented with respiratory symptoms during the early stages of their illness or required admission to the ICU were found to be at an increased likelihood of developing Long COVID [74,75,76,77].


The current understanding of Long COVID risk factors remains incomplete. Further research using large, diverse samples and comprehensive pre-pandemic characteristic measures is necessary to enhance our understanding and develop evidence-based strategies for intervention and service planning [77]. Ongoing research aims to pinpoint specific factors contributing to the risk of developing Long COVID and shed light on the underlying causes behind the variability in symptoms observed among patients.

## 5. Racial and Ethnic Disparities in Long COVID

Health disparities, characterized by unequal health outcomes and limited healthcare access among different population groups, particularly affect racial and ethnic minorities and those with disabilities [78,79]. These disparities stem from various factors such as socioeconomic status, discrimination, systemic racism, and limited healthcare access, with significant implications for the incidence, severity, and long-term consequences of COVID-19 and the associated Long COVID [15,22].

A retrospective study conducted in Louisiana, USA, early in the COVID-19 pandemic in 2020 revealed that 76.9% of hospitalized COVID-19 patients and 70.6% of those who died were identified as black [80]. However, the study did not find a significant association between the black race and higher in-hospital mortality, suggesting that other factors may influence mortality rates, and similar findings were reported by Mackey et al. [78].

On the contrary, a study by Pan et al. showed that both black and Hispanic populations had a higher likelihood of infection and hospitalization, but there were no subsequent differences in hospital treatment or in-hospital mortality. These findings support the hypothesis that the overall greater burden of the COVID-19 pandemic among racial and ethnic minorities could be attributed to a heightened susceptibility to contracting the SARS-CoV-2 virus, influenced by adverse social determinants of health within these minority communities [81].

Long COVID and Racial Disparities:

The impact of race on Long COVID extends beyond the acute phase of the illness. Emerging evidence indicates that racial and ethnic minorities are more likely to experience persistent symptoms and prolonged recovery than white patients [9,79,82]. Marginalized communities face additional challenges in accessing specialized care, rehabilitation services, and mental health support, exacerbating health disparities and increasing the burden on these communities.

Global Impact:

The impact of race on Long COVID is not limited to specific regions but is a global issue. Racial and ethnic disparities in healthcare access and outcomes exist in various countries, influencing the trajectory of Long COVID for marginalized populations worldwide. Research conducted in Latin American countries such as Colombia and Brazil has shown a high prevalence of Long COVID symptoms [83,84]. Similar findings have been observed in South Africa [7], highlighting the need for post-acute care services in resource-constrained settings where physical, cognitive, and mental health disabilities are often overlooked.

Additionally, these countries generally lack robust social safety nets, and the potential impact of long-term consequences on the workforce and families’ livelihoods remains a cause for concern [7].

Conclusion:

In summary, health disparities related to COVID-19 and Long COVID, particularly impacting racial and ethnic minorities, are influenced by complex factors such as socioeconomic status, discrimination, and limited healthcare access. These disparities are not confined to specific regions and have global implications, underscoring the need for inclusive healthcare strategies and support for marginalized communities worldwide.

## 6. Diagnosis of Long COVID

Diagnosing Long COVID can be challenging because distinguishing between complications from COVID-19, treatment-related effects, and other health issues can blur the diagnosis. In addition, patients may suffer treatment-related complications or adverse effects, complications from COVID-19, such as pneumothorax or thromboembolic events such as deep vein thrombosis, pulmonary embolism, stroke, and psychosocial issues. Thus, it may be difficult to differentiate which symptoms are due to Long COVID [85]. In addition, in patients suspected of having acute COVID-19 infection but with a negative PCR test, diagnosing Long COVID can be even more problematic. To date, only one proposed diagnostic criteria for Long COVID has been published [86]. Also, these criteria consist of three sectors: essential, clinical, and duration criteria. They categorize symptomatic and asymptomatic cases during the acute phase as confirmed, probable, possible, or doubtful based on various tests and community prevalence. Diagnosis requires more than 2 weeks for mild, more than 4 weeks for moderate to severe, and more than 6 weeks for critically ill cases in the acute phase. Asymptomatic cases can be diagnosed if symptoms appear 1 week after positive antibody or 2 weeks after positive tests or exposure to COVID-19. Doubtful cases meet Long COVID criteria once symptoms emerge. Thaweethai et al. conducted a prospective survey-based cohort study involving adult patients infected and uninfected with SARS-CoV-2 [86]. The study aimed to ascertain patient-reported symptoms and developed a data-driven scoring framework to classify Long COVID, referred to as Post-Acute Sequelae of SARS-CoV-2 (PASC) as a condition specific to SARS-CoV-2 infection. Higher PASC scores were associated with increasingly poor well-being and functioning measures. Although the PASC score was based on 12 specific symptoms, other symptoms also showed significant correlations with this subgroup, considering their potential adverse impact on health-related quality of life [86].

## 7. Management and Treatment Approaches for Long COVID

### 7.1. Clinical Evaluation of Patients with Long COVID

The management approach of patients with this disorder requires a comprehensive physical, cognitive, and psychological assessment, given the multidimensional aspect of this disease. The current guidelines recommend scheduling the visit in the fourth week after the COVID diagnosis [87,88]. Gathering a detailed medical history with an emphasis on the patient’s comorbidities and the details of the COVID-19 illness supported by a thorough physical examination is a must to check for the presence of complications related to the primary COVID infection and to help rule out other differential diagnoses and evaluate the extent and severity of the patient’s symptoms.

### 7.2. The Recovery Period for Patients with Long COVID

Full recovery from Long COVID symptoms can occur. However, a French study based on a self-reported tool revealed that 85% of patients continued to report persistent symptoms one year after the onset of their symptoms [12]. Variable factors influence the rate of symptom resolution and the time to recovery in patients with Long COVID. Those who suffer from severe COVID-19 infection requiring prolonged hospital admissions or critical care stay tend to require a longer duration for symptom resolution [89]. However, Long COVID symptoms were reported in those with milder disease [90]. Based on a meta-regression and pooling of 54 studies and two medical record databases encompassing 1.2 million people [91], hospitalized patients had Long COVID symptom clusters lasting for 9.0 months compared to 4 months in non-hospitalized patients. Furthermore, comorbidities and advanced age are associated with a longer recovery course [92].

### 7.3. Diagnistic Work-Up of Patients with Long COVID

There are no specific recommendations regarding the required investigations for suspected Long COVID-19 [93]. Standard laboratory evaluations, such as hematological profile with blood counts, complete metabolic panel, endocrine assessment for thyroid disease or diabetes, and investigations into vitamins or iron deficiency, are often performed. However, these evaluations are usually unremarkable [88,93]. The available guidelines encourage an individualized approach to targeted laboratory and radiological investigations and specialty consultation so patients with cardiac, pulmonary, and neurologic, etc. can be evaluated with appropriate investigations according to their symptoms [93]. Obtaining SARS-CoV-2 serology is recommended only for those with no prior documented positive testing.

### 7.4. Treatment of Long COVID

There are currently no broadly effective treatments for Long COVID, and the management is based on symptom-specific pharmacological options. Guidelines for managing long COVID are mostly rapid practical recommendations rather than evidence-based guidelines [88,94]. The European Society of Clinical Microbiology and Infectious Diseases (ESCMID) has developed evidence-based guidelines to guide the evaluation and treatment of patients with Long COVID. However, despite these guidelines, the available evidence is currently inadequate to offer specific recommendations, except for conditional guidance [95] implying uncertainty due to low evidence and thus, recognizing patient preferences is vital in clinical decision-making [93].

Various interventions have been explored for managing fatigue, including complementary and alternative medicine such as acupuncture, cupping, counseling, cognitive-behavioral therapy, and exercise [96]. However, the studies on this topic have shown heterogeneity, and the evidence is inadequate to recommend any particular intervention [93]. Despite not receiving approval from the U.S. Food and Drug Administration (FDA) for the treatment of chronic fatigue syndrome/Myalgic Encephalomyelitis, Rintatolimod, a restricted Toll-Like Receptor 3 agonist, has recently obtained FDA clearance to conduct a clinical trial specifically targeting Long COVID (NCT05592418) [97].

A trial conducted on 60 patients diagnosed with Long COVID demonstrated a potential benefit when administering selective serotonin reuptake inhibitors (SSRIs) [98]. Recent research has highlighted the potential of clomipramine as a medication for managing symptoms associated with Long COVID. Clomipramine is anti-inflammatory and can efficiently cross the blood–brain barrier (BBB) [99]. This makes it an attractive candidate for addressing the specific neurological aspects of Long COVID. Furthermore, melatonin has been acknowledged for restoring disrupted circadian rhythms, a common occurrence in patients with depression [100]. By regulating circadian rhythm, melatonin may hold promise as an adjunct therapy for managing depression symptoms in long COVID patients. However, further extensive studies are required to fully understand and validate the potential benefits of clomipramine and melatonin in the context of long COVID [101]. A study involving 52 patients with Long COVID investigated the use of Low Dose Naltrexone. It determined that it was safe and potentially beneficial in improving overall well-being and reducing symptoms [99].

Antihistamines have been suggested to counteract abnormal mast cell activation and subsequent inflammatory responses and suppress viral growth [100]. Observational studies have indicated that antihistamines can alleviate symptoms related to long COVID [102]. However, additional clinical trials and research are necessary to explore and identify the potential use of antihistamines in managing COVID-19.

### 7.5. Self-Management for Long COVID

Due to limited treatment options and restricted healthcare access during lockdowns, patients with Long COVID turn to self-prescribed modalities, including over-the-counter medications, remedies, supplements, therapies, and dietary modifications [103]. However, self-prescription poses risks such as drug interactions, inappropriate treatments, and high costs and patients are advised to consult their healthcare professionals before pursuing self-management. In response to the need for comprehending this public health concern that has received limited research attention, the therapies for Long COVID Study (ISRCTN15674970) were initiated and will evaluate self-practices among non-hospitalized patients [104].

### 7.6. Rehabilitation Programs

Long COVID can be debilitating, leading to an impaired quality of life. It has been seen that patients often require multidisciplinary care involving continued monitoring of ongoing symptoms, ientification of potential complications for timely intervention, and physical rehabilitation as well as social support [105]. Physical therapy, cardio-pulmonary rehabilitation, nutritional supplements, and olfactory training have been suggested as potential interventions for patients with Long COVID. However, a recent systematic review highlighted the limited evidence supporting their effectiveness in reducing fatigue, dyspnea, improving physical capacity, and enhancing quality of life. The available studies mainly consist of limited randomized controlled trials and cohort studies [93,106].

### 7.7. Prevention of Long COVID

Standard techniques (including masking, social isolation, hand hygiene, and vaccination) to prevent the transmission of COVID-19 are the most effective way to prevent long-COVID. Reinfection contributes additional risks to long-term COVID [107]; hence, vaccination was shown to lower the rates of Long COVID symptoms [108]. Antiviral medications such as nirmatrelvir, molnupiravir, and ensitrelvir may be useful in lowering risk of all-cause hospitalization and emergency department visits compared with untreated patients [109,110]. Metformin might also reduce the risk of Long COVID as it was found to exhibit in vitro activity against SARS-CoV-2 in physiologically relevant doses in cell culture and human lung tissue [111].

## 8. The Economic Ramifications of Long COVID Worldwide

The COVID-19 pandemic has left an indelible mark on global economies, precipitating widespread unemployment, business closures, and a decline in the gross domestic product (GDP). Consequently, pressing economic questions have emerged, such as whether the financial costs of policy interventions to curb the virus are justified by potential health benefits [112,113]. Additionally, each premature death attributed to Long COVID carries substantial societal financial repercussions, prompting attention to various economic concepts during the pandemic to unravel the complexities of these billion-dollar questions [114]. One notable concept of interest is the Value of Statistical Life (VSL), also referred to as the “value of a prevented fatality” [112,115]; that does not directly evaluate the inherent value of life but instead measures the monetary value an individual is ready to spend for a small decrease in the probability of death [113]. The estimates indicate that the VSL for an average American stands at approximately $7.2 million [115].

Long COVID affects a significant proportion of patients who recovered from COVID-19, resulting in persistent health issues that can have profound economic consequences [116]. Beyond the acute phase, Long COVID poses persistent health issues with profound economic impacts. The symptoms hinder workforce productivity and lead to financial setbacks for patients and the broader economy [95,116,117]. A notable Swiss survey by the Federal Social Insurance Office reported an increasing percentage of disability insurance claims attributed to the post-COVID-19 condition [95,117].

The impact of Long COVID on the workforce is particularly significant in service jobs, such as healthcare, social care, and retail. The ongoing labor shortage in these sectors has driven up wages and prices, contributing to the recent inflation surge in the US [118]. One of the primary economic consequences of Long COVID lies in its effect on the labor market. Patients with Long COVID often encounter challenges in returning to work or maintaining their pre-illness productivity levels. Some may require extended sick leave or workplace accommodations, leading to reduced work hours, job loss, or decreased earning potential. Consequently, households experience income loss, and there is an increased reliance on social welfare programs, further straining public resources [118,119].

Moreover, Long COVID exacerbates the already substantial healthcare costs associated with the COVID-19 pandemic. The prolonged medical care required to manage Long COVID symptoms, including specialist consultations, diagnostic tests, and rehabilitative services, contributes to higher healthcare expenditures [117,120]. This places an additional burden on healthcare systems, diverting resources that could be allocated to other critical areas, such as preventive measures or addressing other health concerns [120].

In conclusion, Long COVID presents a substantial economic problem, adding to the challenges societies are already facing in the pandemic’s aftermath. The enduring symptoms experienced by recovering COVID-19 patients lead to significant financial losses worldwide, impacting people, businesses, and economies. A comprehensive understanding of the economic repercussions of Long COVID is vital for policymakers, healthcare systems, and businesses to strategize in alleviating the financial burden, assisting those affected, and promoting sustainable recovery in a post-pandemic world.


**Mitigating Economic Repercussions of Long COVID:**
Workplace Accommodations: Employers can offer flexible work hours, remote working options, or role adjustments to accommodate employees suffering from Long COVID symptoms. Such accommodations can reduce absenteeism and prevent the potential loss of experienced staff.Specialized Rehabilitation Programs: Governments and healthcare institutions can establish rehabilitation centers focusing on post-COVID care. Such centers can aid in faster recovery and quicker reintegration into the workforce.Economic Incentives: Governments can provide tax breaks or subsidies to businesses that offer accommodations for Long COVID-affected employees. This could offset potential productivity losses and incentivize more businesses to make the necessary adjustments.Awareness Campaigns: Widespread public awareness campaigns about Long COVID can drive understanding and empathy. This could foster a more supportive environment for patients, both in workplaces and the broader community.Social Welfare Programs: Expanded social welfare programs can provide temporary financial support for those unable to work due to post-COVID symptoms, reducing the long-term economic strain on households.Research and Development: Governments and private entities can invest in research to better understand Long COVID, which could lead to effective treatments and therapies, thereby reducing its economic impact.


## 9. Conclusions and Future Directions

The COVID-19 pandemic has resulted in a significant number of patients recovering from SARS-CoV-2 infection, with a considerable proportion experiencing Long COVID. Further research using large, representative samples and comprehensive pre-pandemic measures is needed to improve our understanding of Long COVID syndrome’s risk factors and prevalence. Ongoing research aims to identify specific factors contributing to its development and unravel the variability in symptoms observed among patients. Diagnosing Long COVID can be challenging due to the multifactorial nature of the symptoms, and treatment options are limited. Clinical evaluation, tailored investigations, and specialized consultations are necessary to manage Long COVID effectively. Rehabilitation programs and self-management techniques may offer some relief to patients experiencing long-term symptoms.

As research on Long COVID continues, it is essential to understand the underlying pathophysiology, risk factors, and treatment strategies better. Comprehensive and evidence-based guidelines are necessary to assist healthcare professionals in providing optimal care for Long COVID patients. Preventive measures, such as vaccination and antiviral therapy during the acute phase of COVID-19, may help reduce the risk of developing Long COVID. By addressing the challenges posed by Long COVID, societies can work towards mitigating the health disparities and economic burden associated with this complex condition.

The intersection of race, health disparities, and Long COVID syndrome underlines the urgency of addressing systemic inequities in healthcare globally. Racial and ethnic minorities bear disproportionate burdens related to COVID-19 and its long-term consequences, perpetuating existing disparities. These disparities stem from various factors such as socioeconomic status, discrimination, systemic racism, and limited healthcare access, with significant implications for the incidence, severity, and long-term consequences of COVID-19 and the associated Long COVID. Acknowledging and addressing these disparities can lead to equitable access to care, support, and resources, thereby mitigating the impact of Long COVID syndrome on marginalized communities worldwide.

A comprehensive approach is required to address health disparities and the influence of race on Long COVID syndrome. Policymakers and healthcare systems should prioritize equity in healthcare delivery, ensuring equal access to testing, treatment, and post-acute care services. Efforts should concentrate on increasing health literacy, community engagement, and implementing tailored interventions for marginalized populations. Collecting disaggregated data on race and ethnicity is vital in identifying and monitoring disparities and guiding targeted interventions to promote health equity.

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
