# Peer review of "From Acute Infection to Prolonged Health Consequences: Understanding Health Disparities and Economic Implications in Long COVID Worldwide"

_ijerph, 2024, doi:10.3390/ijerph21030325_

Round 1

Reviewer 1 Report (Previous Reviewer 1)

Comments and Suggestions for Authors

 Now it can be accepted 

Author Response

Thank you very much for taking the time to review the manuscript for the second time. We highly appreciate your thoughtful comments and suggestions that helped us improve our submitted manuscript.

Reviewer 2 Report (Previous Reviewer 2)

Comments and Suggestions for Authors

Dear authors,

I do not have much to add because your work seems complete. Perhaps you could do the following:

The introduction (just background) is acceptable.

Subsection B (literature search) should be a second section on its own, called "Methodology" or "Material and method." Regarding this section, just telling established guidelines is not enough. Guidelines by whom? Are they a standard?

The keywords seem fine. How did you filter the articles for your review? Did you include review articles? Book chapters? In other words, which inclusion or exclusion criteria did you use? How many eligible documents did you find? Did you use some software or special tools for your review? Were there constraints or limitations? How did you overcome it?

The following source will help you improve your methodology: https://www.sciencedirect.com/topics/psychology/narrative-review

Besides that, I think your manuscript is acceptable.

Author Response

We appreciate the respectful reviewer's emphasis on the literature search methodology utilized for this review article. As stated in our response to the previous round of revisions, our literature search methodology involved a targeted exploration of published articles focusing on long COVID-19, with a cutoff date of July 30th, 2023. We provided the keywords used and mentioned the online databases that were searched. No software was used for literature extraction, and no inclusion or exclusion criteria were applied since this was not a systematic review. Review articles, original studies, and case reports were included, while book chapters were not considered. Article selection was at the discretion of the authors responsible for writing the assigned subsections based on their expertise. The reference we utilized for guidelines on writing narrative reviews (Gasparyan, Armen Yuri, et al. "Writing a narrative biomedical review: considerations for authors, peer reviewers, and editors." Rheumatology international 31 (2011): 1409-1417) has been cited 599 times and served as a helpful resource in crafting this review.

Regarding the suggested source mentioned by the responding reviewer, our method aligned with the suggested outline as the source clearly stated: "Summarize and synthesize the findings from the articles you have found, and integrate them into your writing as appropriate. You do not need to document your literature search. Reference the articles as you use information from the studies.””

So, in agreement with the suggested comments, we made the search methodology a separate subsection and became subsection number 2, and the subsequent subsections were renumbered accordingly. The methods section reads now as follows:

Following published guidance on writing narrative reviews (10), our literature search methodology involved a targeted exploration of published articles focusing on long COVID-19, with a cutoff date of July 30th, 2023. The keywords utilized included “COVID”, "Long COVID," "post-COVID conditions," "post-acute sequelae of SARS-CoV-2 infection," "post-acute COVID-19," "chronic COVID-19," and "post-COVID syndrome." Review articles, original studies, and case reports were included, while book chapters were not considered. Article selection was at the discretion of the authors responsible for writing the assigned subsections based on their expertise.We searched across diverse online platforms, including MEDLINE, Scopus, Web of Science, and the Google Scholar search engine, ensuring a comprehensive data collection.  The articles used in writing this review were cited, and the full list of references is available at the end of the manuscript.

Reviewer 3 Report (Previous Reviewer 3)

Comments and Suggestions for Authors

I believe that the article has been substantially improved so I believe it could be published.

Author Response

Thank you very much for taking the time to review the manuscript for the second time. We highly appreciate your thoughtful comments and suggestions that helped us improve our submitted manuscript.

This manuscript is a resubmission of an earlier submission. The following is a list of the peer review reports and author responses from that submission.

Round 1

Reviewer 1 Report

Comments and Suggestions for Authors

The review article aims to investigate the relationship between race, health disparities, and long COVID, highlighting the disparities faced by marginalized communities worldwide. The study is worthy, and whole manuscript was well written, however many points need to be considered. I think the abstract is too long while the conclusion needs more improvement. 

Reviewer 2 Report

Comments and Suggestions for Authors

Please find below my comments and suggestions for improvements:

Title:

The title is comprehensive and encapsulates the central theme of the paper. It mentions health disparities and economic implications, signaling the reader what to expect.

Abstract:
The abstract provides a good overview of the article's main themes. However, a few points to improve readability and coherence:

  1. "Addressing health disparities and mitigating the economic impact of Long COVID require a comprehensive approach focused on equity in healthcare access and targeted interventions." – This sentence could be more concise, e.g., "Mitigating Long COVID's economic impact requires a focus on healthcare equity and targeted interventions."
  2. "Managing of long COVID involves evaluating..." – Consider rephrasing to: "Managing long COVID involves..."
  3. "Managing long COVID involves evaluating and addressing the physical, cognitive, and psychological symptoms" seems repetitive, given that a similar point is made in the following sentence about management.
  4. The last few lines focus more on the economic side. It might be worth bringing some structure to the abstract by separating health and economic effects.
  5. The abstract could be improved by summarizing the main findings or insights of the paper at the end.

Keywords:
Keywords are apt for the subject. Ensure they are the terms most relevant to the main focus of the article and its intended audience.

General Observations:

  • The abstract thoroughly discusses Long COVID's medical aspects and symptoms. It would be beneficial to include a brief overview of the critical economic insights, given that the title and the broader scope of the paper focus on financial implications.
  • It is crucial to ensure consistency in terms – sometimes you use "Long COVID," other times "long COVID." Consistency in terminology will aid reader comprehension.
  • Ensure that line numbers are consistent and sequential. There seems to be a repeated use of some line numbers, which can confuse during the review process.
  • Consider whether "prevention involves standard techniques and the use of antiviral medications and metformin" provides enough context in the abstract. It may raise questions about what "standard techniques" entail.

In general, the abstract looks comprehensive and relevant. It seems to be a well-structured paper discussing Long COVID's health and economic aspects. Minor revisions for clarity and coherence will elevate the manuscript.

 Introduction

Content & Structure:
The introduction is comprehensive, from the basic understanding of Long COVID to discussing the nuances, its connection with health disparities, and the importance of economic implications.

Specific Feedback:

  1. Sentence Structure & Repetition:
    • "The COVID-19 pandemic has resulted in a growing number of individuals recovering from acute infection with persistent lingering symptoms." Consider shortening this to: "Many individuals recovering from COVID-19 continue to experience lingering symptoms."
    • Avoid repeating "The COVID-19 pandemic" (Lines 57 and 77). The second mention could begin as, "Beyond the immediate health impacts of the virus..."
  2. Citation & Terminology Consistency:
    • Ensure all statements are appropriately cited. You have done an excellent job here, but double-check that none of the statements require additional references.
    • Decide on a consistent style for citing references. For instance, "(1)" vs "(1-3)". The first is used at the start, while the latter is used later in the introduction.
  3. Clarity:
    • The myriad terms used for Long COVID are mentioned; this is good for clarity. After listing the words, the statement "For this review, the term Long COVID will be used" is excellent for setting expectations.
    • Consider revising: "The COVID-19 pandemic has not only exposed the severe health impacts of the virus but has also brought attention to existing health inequalities among different racial and ethnic groups." to "The pandemic has highlighted both the severe health impacts of the virus and existing health disparities among different racial and ethnic groups."
  4. Focus:
    • The introduction's scope is broad; ensure that all points mentioned (clinical manifestations, diagnostic criteria, pathophysiology, organ consequences, complications, risk factors, management, treatment, economic ramifications, and health disparities) are adequately addressed in the manuscript.
  5. Paragraph Division:
    • Consider dividing the introduction into smaller paragraphs for readability. A potential break can be after a "comprehensive understanding of Long COVID."
  6. Endnote:
    • Setting the reader's expectations toward the end of the introduction is a good practice. You have done this by mentioning the review's scope, which is excellent. Ensure the body of the paper addresses each of these points.

The introduction provides a clear foundation for the reader, setting up the importance of the topic and highlighting the review's scope. Minor improvements in sentence structure and clarity can further enhance its effectiveness.

2. Clinical Manifestations of Long COVID: Immunological and Pathophysiological Mechanisms

  • The definitions and prevalence of symptoms were mentioned. However, using bullet points for the commonly listed symptoms (Fatigue, respiratory symptoms, difficulty sleeping, etc.) would make it more readable.
  • Some numbers are provided in parentheses to cite sources, but consistently mentioning what those numbers represent can clarify whether they are references or footnotes.
  • Instead of merely stating the various mechanisms, explaining briefly how each directly results in the described symptoms can offer clarity.

2.1. Musculoskeletal system involvement

  • Using whitespace or bullet points could simplify the presentation of information, especially when describing different research findings.
  • The term "long COVID" is used inconsistently in the text; it should be standardized (e.g., "Long COVID" vs. "long COVID").

2.2. Respiratory system

  • You can group symptoms, complications, and clinical findings to enhance readability. Use subheadings or bullet points for this purpose.
  • When introducing terminology or abbreviations like PFT, FEV1, FVC, or TLC, ensuring they are clearly defined for readers unfamiliar with the jargon can be helpful.

2.3. Cardiovascular system

  • This section provides an extensive list of complications and findings but could be improved by linking the observations directly to the underlying mechanisms.
  • It might be helpful to differentiate between findings from observational studies and mechanisms visually. Subheadings, bullet points, or visual aids can be beneficial here.

2.4. Nervous system

  • "Neuro-COVID" or "Post-COVID-19 Neurologic Syndrome" is introduced without defining the difference or indicating if they are synonymous. Clarification is required.
  • When discussing the PET scans, the importance of the "100% correct classification" is noted, but some context or comparison to standard measurements would make the finding's significance more straightforward.

2.5. Renal system involvement

  • Given its importance, explaining the significance of the ACE2 receptor in layperson's terms early in this subsection would be beneficial.
  • The subsection introduces several mechanisms causing kidney injury post-COVID but might benefit from explaining why this matters for long-term patient health or prognosis.
  • Consideration should be given to the flow of the content, ensuring that concepts are introduced before diving deep into the specifics (e.g., COVAN is detailed, but a brief overview before the deep dive would help).

2.6. Gastrointestinal system involvement

  1. Consistency: Uniform the way you are reporting percentages. If you are using "approximately 12%," try to avoid "around 42%" in the psychological sequelae section and use "approximately" for consistency.
  2. Elaborate on the Gut Microbiome: The section that discusses the gut microbiome could be expanded to provide more context about the relevance and importance of the gut microbiome in human health.
  3. Clarity: Instead of "GI manifestations," "gastrointestinal manifestations" might be more evident for readers unfamiliar with the abbreviation.

2.7. Endocrine system involvement

  1. Clarity and Abbreviations: Not everyone might understand the abbreviation "T2DM" for type 2 diabetes mellitus. Either expand it or explain it the first time you use it.
  2. Uniformity in Citing Studies: If using the format "Y Xie and Al-Aly," try to maintain that format throughout rather than just citing numbers.

2.8. Psychological symptoms/sequelae

  1. Whitespace: Avoid too much space between lines or references, like between lines 261 and 262.
  2. Consistency with Other Coronaviruses: When mentioning SARS and MERS, it would be helpful to specify them as Severe Acute Respiratory Syndrome and Middle East Respiratory Syndrome, respectively, at least on the first mention.

2.9. Cutaneous symptoms/sequelae

  1. Explain Dermatological Terms: The terms "morbilliform," "pernio-like," "urticarial," etc., might not be familiar to all readers. Consider providing brief explanations or parentheticals for clarity.
  2. Consistency in Description: There is a mention of "laboratory-confirmed COVID-19 patients," but how this group differs from the other COVID-19 patients mentioned earlier in the text is unclear. Maintain clarity when distinguishing between patient groups.
  3. Conclusion for Each Subsection: It might be beneficial to conclude each subsection with a summary statement or potential implications for future research or clinical practice, providing readers with a clear takeaway from each section.

General Notes for Section 2:

  • While the text does an excellent job presenting scientific information, it may benefit from providing context or implications for each finding. This would help with comprehension for a lay audience and emphasize the importance of each result.
  • Consider using tables, figures, or infographics to summarize key points, especially when comparing studies or presenting a broad range of data.
  • Formatting: Be consistent with line spacing and indentation. Some sections seem more spaced than others.
  • Consistent Citations: It might be easier for the reader if all citations were either at the end of sentences/ideas or immediately after the fact they support, rather than in the middle of sentences.
  • Language Uniformity: For a formal scientific document, it is essential to maintain a consistent tone and style throughout.

 3. Risk factors and Demographics leading to increased susceptibility to Long COVID

  • Structure: Consider utilizing bullet points or tables to make the presentation of risk factors more concise and digestible.
  • Redundancy: Phrases like "emerging evidence indicates" could be more concise. Instead, state the facts directly.
  • Citations: The citation formatting seems inconsistent. In some places, references are noted with multiple sequential citations (e.g., "female sex(67–70)"), and in others, they are interspersed.
  • Clarification: The statement "a shorter hospital stay was inversely associated with the syndrome" might benefit from a brief explanation.

4. Understanding Health Disparities and Racial Implications in Long COVID

  • Title: The title could be more concise: "Racial and Ethnic Disparities in Long COVID."
  • Clarity: The section discusses various studies with different results. Highlighting the main takeaways from each can make the text more understandable.
  • Global perspective: When discussing the global impact, organizing findings by region or country would be helpful to make comparisons easier for the reader.
  • Conclusion: Consider summarizing the section's findings at the end to tie everything together.

5. Diagnosis of Long COVID

  • Clarity: Simplify and clarify the challenges of diagnosing Long COVID. For example: "Distinguishing between complications from COVID-19, treatment-related effects, and other health issues can blur the diagnosis."
  • Diagnostic criteria: Elaborate more on the proposed diagnostic criteria for Long COVID. Discuss its components and how it contrasts with other conditions.
  • Terminology: Maintain consistency in terms. For instance, choose between "Long COVID" and "Post-Acute Sequelae of SARS-CoV-2 (PASC)" and stick to it throughout the text. If both terms are used interchangeably, clarify this at the beginning.

 6.1. Clinical evaluation of patients with long COVID-19

  • Clarity: While it is mentioned that an assessment should occur in the fourth week post-diagnosis, it is unclear if this is the first or follow-up evaluation.
  • Depth: The significance of evaluating in the fourth week should be elaborated upon.

6.2. The recovery period for individuals with Long COVID

  • Consistency: The term "Long COVID" is sometimes capitalized. Ensure consistent use.
  • Precision: Define what "symptom clusters" mean or provide a brief explanation.

6.3. Workup for patients with long COVID-19

  • Formatting: Space between 'COVID' and '19'.
  • Precision: For SARS-CoV-2 serology, explain why it is only recommended for those without prior positive testing.

6.4. Treatment of Long COVID

  • Detail: The description of Rintatolimod's mechanism is quite technical. Consider simplifying or providing a brief explanation for lay readers.
  • Clarity: "conditional guidance" - Define or explain what this entails.

6.5. Self-management for long COVID

  • Perspective: Consider the importance of patients consulting their healthcare professionals before pursuing self-management.

6.6. Rehabilitation programs

  • Detail: Provide more context on why rehabilitation is essential, perhaps by highlighting the types of debilitations faced by long-term COVID patients.

6.7. Prevention Of long-COVID

  • Consistency: Maintain consistent capitalization for "long-COVID."
  • Clarity: Clarify the role of medications like Metformin in reducing the risk.

7. The Economic Ramifications of Long COVID Worldwide

  • Depth: Consider including potential strategies or interventions that countries/businesses can adopt to mitigate the economic repercussions of Long COVID.

8. Conclusion and future directions

  • Flow: There is an abrupt jump from discussing Long COVID specifics to discussing racial and ethnic disparities. Consider introducing this shift more fluidly.
  • Completeness: The section about racial and ethnic minorities seems truncated. Please elaborate on the impact on these groups and why they face a disproportionate burden.

 Yours sincerely

Comments on the Quality of English Language

Dear authors,

Please observe the following:

  • Flow: Ensure a logical flow from one section to the next. Each section should smoothly transition into the next topic.
  • Consistency: Make sure that terms, definitions, and abbreviations are consistently used throughout the text.
  • Grammar: Check for minor grammatical errors, such as the placement of punctuation before and after citations.
  • Formatting: Ensure consistent formatting, especially for references. Uniformity makes the text appear more professional and more accessible for readers to follow.

Especially sections 3 to 7.

Yours sincerely

Reviewer 3 Report

Comments and Suggestions for Authors

I would like to thank you for the effort of writing this article, although I think some changes are needed to improve the quality of the information you are trying to convey.

It is necessary to include information in the methods about:  Type of study, number of studies reviewed, selection criteria for selecting the articles, type of language of the articles reviewed.

Please be careful throughout the document when using some abbreviations/acronyms such as ACE2 or CD4, you need to include the abbreviations the first time you use these words (please see lines 105-116 on page 3), UCI (lines 151-152 on page 4), APOL1 (line 230 on page 5), T2DM (line 250 on page 6), please check throughout the document for correct use of acronyms.

Finally I think it is necessary to include a conclusion that reads. It is necessary to differentiate between Long COVID and the consequences of COVID disease in some specific patients, in order to focus treatment and further follow-up.
